# Gatekeepers of the Gut: The Roles of Proteasomes at the Gastrointestinal Barrier

**DOI:** 10.3390/biom11070989

**Published:** 2021-07-05

**Authors:** Gayatree Mohapatra, Avital Eisenberg-Lerner, Yifat Merbl

**Affiliations:** Department of Immunology, Weizmann Institute of Science, Rehovot 7610001, Israel; gayatree.mohapatra@weizmann.ac.il (G.M.); avital.eisenberg@weizmann.ac.il (A.E.-L.)

**Keywords:** proteostasis, ubiquitin–proteasome system, degradation, gastrointestinal barrier, colitis, colorectal cancer, inflammatory bowel disease, auto-immune disease

## Abstract

The gut epithelial barrier provides the first line of defense protecting the internal milieu from the environment. To circumvent the exposure to constant challenges such as pathogenic infections and commensal bacteria, epithelial and immune cells at the gut barrier require rapid and efficient means to dynamically sense and respond to stimuli. Numerous studies have highlighted the importance of proteolysis in maintaining homeostasis and adapting to the dynamic changes of the conditions in the gut environment. Primarily, proteolytic activities that are involved in immune regulation and inflammation have been examined in the context of the lysosome and inflammasome activation. Yet, the key to cellular and tissue proteostasis is the ubiquitin–proteasome system, which tightly regulates fundamental aspects of inflammatory signaling and protein quality control to provide rapid responses and protect from the accumulation of proteotoxic damage. In this review, we discuss proteasome-dependent regulation of the gut and highlight the pathophysiological consequences of the disarray of proteasomal control in the gut, in the context of aberrant inflammatory disorders and tumorigenesis.

## 1. Introduction

Tissues at the gastrointestinal barrier face a constantly changing environment that encounters a multitude of stress signals and antigens from food and [1,2,3,4,5,6]. Adding to the complex milieu of the gut, a plethora of commensal bacteria colonize intestinal tissues, acting as a source for bacterial-derived trigger molecules. The gut barrier, therefore, copes with a critical challenge to continuously sense and discrete between endogenous/symbiotic signals to such derived from pathogens to properly activate inflammatory responses upon need. While a prompt immune response is a key to fighting pathogenic threats, instigating immune activation too soon, or in an abrupt manner, may result in responding to the many ‘false alarms’ derived from non-pathogenic or self-signals, and induce aberrant inflammation. Timely resolution of inflammatory signals is also required to prevent excessive, or chronic, damage. To enable rapid response while controlling the deleterious potential of aberrant immune activation, inflammatory pathways are typically restricted by the constitutive expression of negative regulators. In such a manner, positive mediators of inflammation are held ready for action, yet confined by a threshold required for relief of inhibition and pathway activation. In addition, various protective mechanisms operate at the intestinal epithelial barrier to confer its tolerogenic properties, including toll-like receptor (TLR) signaling, hyperexpression of MHC molecules, scavenging of bacterial- or food-derived antigens by intraepithelial lymphocytes, and presentation of peptides by tissue-resident dendritic cells [7]. Thus, an intricate regulatory network of host immune signaling and interactions is key for controlling and maintaining homeostasis at the gut. Yet, if regulatory mechanisms are breached, threatening conditions such as infection, inflammatory tissue damage, or cancer may arise [8,9,10,11]. Such diverse pathological outcomes require the consideration of different aspects of immune regulation: How do pathogens manipulate host regulators to evade immune responses? Is there a general threshold of activation against infection? How is the susceptibility to autoimmune disease or cancer affected by perturbation of this network? As well as, importantly, how can controlled regulation of proteolytic events, via lysosome or proteasomes, be harnessed to limit pathological outcomes?

To allude to the abovementioned questions, numerous studies investigated the involvement of the main inflammatory pathways such as those of NFκB and inflammasomes, in gut response to pathogens, and aberrant activation in inflammatory bowel pathology [12,13,14]. Yet, underlying many of these processes are upstream regulation by the ubiquitin–proteasome system (UPS), which is far less studied in this context.

The UPS provides rapid and specific response to stimuli at the gut through the modification and targeting to degradation of key inflammatory effectors and regulators (see Box 1: The Ubiquitin Proteasome System, and Figure 1). Further, ubiquitin-dependent proteasomal degradation is implicated in the modulation of inflammatory responses to self and non-self-signals [15,16,17,18,19,20]. In particular, proteasomes contribute to the first line of defense at the gut by regulation of the NFκB pathway [21,22]. Under basal conditions, NFκB is restrained in the cytosol by interacting with the inhibitory protein IκBα. Binding of microbial ligands to pattern recognition receptors (PRR), or exposure to inflammatory cytokines such as TNF-α and IL-1β, activate the IκB kinase (IKK), which induces the phosphorylation of IκBα. This signals for proteasomal degradation of IκBα, relieving its inhibitory interaction with NFκB. Activated NFκB then translocates to the nucleus where it induces the transcription of genes related to immunity, inflammation, and apoptotic cell death, such as NLRP3, pro-IL1β and pro-IL-18, adhesion molecule ICAM-1, antiapoptotic IAPs, FLIP, and Bfl1 [23] (Figure 2A). While inflammation may be induced as a cytoprotective mechanism, for example during ulcer healing, extensive inflammation may induce tissue damage. As such, modulation of proteasome-dependent NFκB activation is crucial also in balancing gut inflammation, prohibiting auto-inflammation, and maintaining gut homeostasis (Figure 3).

Box 1The ubiquitin–proteasome system.The ubiquitin–proteasome system (UPS) plays a pivotal role in protein homoeostasis via an intricate network of enzymes and regulators that together with cellular proteasomes orchestrate tight regulation of protein function and degradation [24,25,26]. Among the multitude of functions regulated by the UPS are processes such as cell cycle, gene transcription, cell survival, apoptosis, epithelial barrier function, inflammation, myocardial function, or lung function [27,28,29,30,31,32]. Further, proteasome-dependent degradation generates peptides for antigen presentation on MHC I, which are crucial for adaptive immunity, maintaining of self-immune tolerance, and eradication of cancer cells [33,34,35].Protein modification by ubiquitin requires sequential activity of three enzymes, namely, an E1-activating enzyme, an E2-conjugating enzyme, and a ubiquitin E3 ligase [36]. This ATP-dependent enzymatic cascade results in the conjugation of ubiquitin onto a substrate protein, either as a single 76 amino acid polypeptide (mono-ubiquitination) or a chain of ubiquitin molecules (polyubiquitination). Deubiquitination refers to the process by which ubiquitin molecule is removed from the target protein by the proteolytic activity of deubiquitinating enzymes (DUBs). Thus, DUBs facilitate the generation of free Ub molecules for further conjugation, trimming of ubiquitin chains and reversal of ubiquitination signaling by removal of Ub from target proteins (Figure 1). Ubiquitin itself contains several lysine residues (K6, K11, K27, K29, K48, and K63) or the *N*-terminal methionine residue (M1), which may be conjugated by other ubiquitin molecules to form linkage-specific chains that induce specific effects on the substrate ranging from activation and promoting complex assembly to targeting substrates to proteasomal degradation [37].Canonical 26S proteasome complexes are assembled by the 20S core particle and the 19S regulatory particle. The 20S core is a barrel-shaped structure of two outer alpha rings and two inner beta rings, each composed of seven structurally similar alpha or beta subunits [38,39,40]. The alpha rings serve as a gate for the entry of proteins into the chamber while the beta rings form the proteolytic chamber. Three beta subunits, β1, β2, and β5, comprise the catalytic core of constitutive proteasomes and perform distinct proteolytic activities including caspase, trypsin, and chymotrypsin-like activity, respectively [26]. Under inflammatory conditions, the expression of immunoproteasome-associated subunits is upregulated. Immunoproteasomes are assembled with different catalytic subunits, namely, β1i (LMP2), β2i (MECL1), and β5i (LMP7) (Figure 1). While the nature of the proteolytic activities of β2 and β5 are not altered by the replacement to their immunoproteasome counterparts, the chymotryptic-like site β1i replaces the caspase activity associated with β1 in constitutive proteasomes. This change in catalytic activity results in altered cleavage properties of immunoproteasomes, and is associated with the established role of immunoproteasomes in promoting the cleavage of hydrophobic residues, considered to be preferential for binding of TAP and the MHC I molecules [41,42,43,44]. While constitutive proteasomes are the predominant form in most eukaryotic cells [26], the immunoproteasome is highly expressed in immune cells and is induced by IFNγ and in various types of cancers [45,46]. The differential expression of proteasome components in distinct tissues highlights their potential roles in tissue-specific functions.

Below, we revisit current knowledge on proteasome-dependent regulation at the gastrointestinal barrier and its involvement in inflammation and colorectal cancer.

## 2. UPS-Dependent Gastrointestinal Inflammation

### 2.1. The Exploitation of Proteasomal Degradation by the Gut Microbiota and Enteric Pathogens

The dependence of NFκB on proteasome-mediated activation is exploited by various bacteria to manipulate the instigation of host inflammation. The commensal microbiota *Lactobacillus* and *Streptococcus thermophilus* were shown to inhibit proteasome activity in gut epithelial cells, resulting in reduced IκB degradation and suppressed NFκB activation [50]. While the molecular mechanism that mediates this phenomenon has not been resolved, it was demonstrated to be mediated via bacterially secreted factors early upon engagement of the bacteria by the gut epithelial cells. While such induction of tolerance towards commensals is symbiotic with the host, it may be deleteriously exploited by pathogens. For example, infection by *Shigella flexneri,* an enteric pathogen that elicits severe inflammatory damage to the gut mucosa, was promoted in the presence of the commensal bacteria *L. casei*, which attenuated the inflammatory response to the infection by modulating the expression of several enzymes of the ubiquitination machinery to reduce IκB degradation ([47], Figure 2B). Another example describing how proteasome-dependent attenuation of inflammation by commensal bacteria may support pathogens was described by Neish et al., who demonstrated that non-virulent *Salmonella* attenuated the inflammatory response to pathogenic *Salmonella Typhimurium* [51]. Specifically, non-virulent *Salmonella*, but not pathogenic *Salmonella Typhimurium*, blocked IκB ubiquitination by the SCFβ-TrCP E3 ligase, and thereby prevented the consequent IκB degradation and NFκB activation. Colonization with non-virulent *Salmonella* also blocked the ubiquitination of β-catenin, another SCFβ-TrCP substrate, but did not globally affect cellular ubiquitination, suggesting that non-virulent *Salmonella* elicits anti-inflammatory effects via selective inhibition of SCFβ-TrCP. In contrast to *L. casei*, which could promote proteasome inhibition via secreted factors, *Salmonella* required direct interaction with the epithelial cells to induce its effect [51].

Certain pathogens stimulate proteasome activity to promote their colonization. For example, infection by adherent-invasive *Escherichia coli* (AIEC) increased proteasome activity, which contributed to bacterial proliferation [20]. While the mechanism is still not fully understood, it was speculated that clearing up ubiquitinated proteins, by increasing proteasome activity, alleviates infection-induced proteotoxic stress that may induce cell death. Another UPS-dependent mechanism by which AIEC colonization and pathogenicity were promoted is via the reduction in expression of the deubiquitinase CYLD [20]. The effect of CYLD on infection and the consequent inflammatory response is complex. CYLD restricts inflammation by inhibiting NFκB activation [52], yet it can induce innate immune responses by stabilizing STING, a cytosolic DNA sensor and a major regulator of type I interferon signaling, via its deubiquitination on K48-linked chains [53]. In addition, CYLD regulates innate immune signaling by deubiquitinating NLRP6, a component of the microbial sensor the inflammasome. Deubiquitination inhibits the complex formation of NLRP6 and ASC and regulates the maturation of IL18, thus critically limiting intestinal inflammation ([48], Figure 2B).

### 2.2. Proteasomes in Inflammatory Bowel Disease

Induction and resolution of inflammation are regulated both temporally and spatially across the gut tissue. Interestingly, proteasomes have been described as double-edged swords that may either promote or restrict colitis, an inflammatory bowel disease (IBD) of the colonic mucosal surface, depending on the stage in which they are activated [54,55,56,57]. Proteasome inhibition by the broad inhibitor MG132 was shown to abrogate the development of spontaneous colitis in IL10−/− mice [58], suggesting that proteasomes are required for initiation of colitis. The suppressive effect of MG132 was mediated by reducing the proliferation of intestinal epithelial cells, thus relieving the hyperplasia that is associated with gut inflammation in IBD. Furthermore, proteasome inhibition was shown to decrease NFκB and TNF-α activation in the colonic tissue in the IL10−/− mouse colitis model [58]. However, examination of dextran sodium sulfate (DSS)- induced experimental colitis, wherein tissue damage and loss of barrier integrity induce the acute response of colitis, revealed that proteasomes were not involved in the initiation of stages, but were rather involved in the NFκB-dependent regeneration of epithelial cells following the injury [58]. Considering the different triggers of colitis in the two systems, one can plausibly speculate that while stimulation of inflammation by proteasomes is essential in the acute response to tissue injury, it may be detrimental in the context of chronic inflammation.

Upon inflammation, the catalytic subunits of the constitutive proteasome are replaced by IFNγ-inducible subunits, leading to the formation of the immunoproteasome [59]. Indeed, immunoproteasomes are expressed in the inflamed mucosa of Crohn’s disease patients, an IBD which involves inflammation along the gastrointestinal tract [60,61]. Nevertheless, several studies suggest that upregulation of immunoproteasomes may be more than merely a reflection of inflammation, but rather a driving factor in the pathophysiology of IBD. First, degradation of IκB and processing of NFκB were shown to be enhanced upon immunoproteasome expression, correlating with increased NFκB inflammatory signaling [60,62]. Further, to understand the early physiological changes during the onset of colitis, Collett et al. used a Mdr1a−/− (multidrug-resistant protein 1a/b) model, wherein mice lack the intestinal transporter P-glycoprotein [63]. Mdr1a−/− mice develop colitis spontaneously upon exposure to normal enteric microflora by the generation of reactive oxygen species. Notably, this study identified the upregulation of the immunoproteasome before the onset of disease, suggesting that immunoproteasome regulation is associated with the pathogenesis of colitis [63]. Additional support for the role of immunoproteasomes in IBD comes from the demonstration that the expression of LMP2, one of the catalytic subunits of the immunoproteasome, was enhanced in colons of DSS-treated mice starting at early stages [64]. To further determine whether immunoproteasomes promote gut inflammation, researchers evaluated symptoms of colitis in mice in which immunoproteasome activity was compromised via genetic deletion of the catalytic subunits or pharmacological inhibition. Indeed, deficiency in LMP2 or LMP7, or block of LMP7 activity, led to a reduction in infiltration of neutrophils and an expansion of Th1 and Th17 cells, suggesting that LMP2 promotes the development of colitis ([64,65,66], Figure 2B). Further, differential activities of the immunoproteasome were associated with irritable bowel syndrome (IBS) and Crohn’s disease. While LMP2 caspase-like activity was reduced both in IBS and Crohn’s disease patients, trypsin-like (MECL1) activity was increased more in IBS compared to Crohn’s disease and control patient samples, and the chymotrypsin-like activity (LMP7) was upregulated in Crohn’s disease patients only [67]. These data suggested that the contribution of different catalytic subunits to disease manifestation may vary, although the mechanisms remain unknown. Nevertheless, since the symptoms of Crohn’s disease and ulcerative colitis are close, it will be intriguing to explore whether the expression of different proteasome subunits may be utilized as a biomarker in molecular profiling of Crohn’s disease and ulcerative colitis. Notably, with the advent of pharmacological agents targeting selective immuno- and constitutive proteasome subunits, a better understanding of proteasome-dependent regulation of gut inflammation may offer novel modalities for therapeutic intervention.

### 2.3. Ubiquitin E3 Ligases in Gut Inflammation

Inflammatory effectors are affected by both genetic alterations and changes in post-translational regulation, such as ubiquitin. Several ubiquitin E3 ligases are implicated in the pathogenesis of IBD through the modulation of key inflammatory regulators such TLRs and inflammasomes.

#### 2.3.1. Inflammasome Activation

The inflammasome is a cellular protein complex that facilitates the production of the inflammatory cytokines interleukin-1β (IL-1β) and IL-18 and induces pyroptotic cell death [68]. Assembly of the inflammasome complex is mediated by NLRP (NOD LRR, and pyrin domain-containing protein) family members upon engagement of exogenous triggers such as pathogen-associated molecular patterns (PAMPs) or danger-associated molecular patterns (DAMPs), including endogenous signals such as reactive oxygen species, mitochondrial DNA, cardiolipin, and HMGB1 [68]. The NLRP3 inflammasome is the most studied inflammasome complex and is prevalent in both the epithelial and immune cells of the gut. Yet, the role of the NLRP3 inflammasome in IBD is controversial, with some studies reporting detrimental outcomes of NLRP3 activation while others suggested cell-protective effects (reviewed in [12,13]). The cellular abundance of NLRP3 is directly controlled by the ubiquitin E3 ligase TRIM31, which promotes K48-linked poly-ubiquitination of NLRP3 and its targeting to proteasomal degradation [56]. Accordingly, deficiency of TRIM31, which would lead to stabilization of NLRP3, was demonstrated to attenuate the severity of DSS- induced colitis, in agreement with the reported protective role for NLRP3 in this model ([56,69], Figure 2B). NLRP3 inflammasomes are further regulated via ubiquitination through the function of the E3 ligase cbl. Cbl ubiquitinates the activated, tyrosine-phosphorylated form of protein tyrosine kinase 2-beta (Pyk2), a critical effector of NLRP3 activation. This targets Pyk2 to degradation and inhibits NLRP3 inflammasome activation [70]. Thus, the Cbl function is important for inhibiting NLRP3 inflammasomes. Cbl is further implicated in NLRP3 inflammasome regulation by maintaining the integrity of mitochondria, which may be a source of mitochondrial DNA, and reactive oxygen species, which may stimulate NLRP3 inflammasomes. Indeed, inhibition of Cbl by the chemical inhibitor hydrocotarnine increased IL-18 secretion and protected from colitis in DSS-treated mice, suggesting that Cbl restrains gut inflammation by limiting inflammasome activation [70].

#### 2.3.2. TLR Activation

The ubiquitin E3 ligase RNF5 induces the degradation of S100A8, an abundant DAMP molecule, which serves as an agonist for TLR4 and drives severe inflammation. RNF5 was shown to restrict colitis in an S100A8-dependent manner, as RNF5 deficient mice demonstrated S100A8 accumulation that induced colitis, which was abrogated by S100A8-neutralizing antibodies [54]. In humans, ulcerative colitis patients have low RNF5 levels. Moreover, RNF5-low/S100A8-high sections of the gut epithelium of colitis patients correlated with severe inflammation [54]. These findings suggest the utility of targeting S100A8 by antibodies or other potential approaches in RNF5-low colitis patients.

Beyond IBD, ubiquitin-dependent proteasomal degradation is critical in neonates, where it is associated with the leading cause of mortality in premature infants, a condition of gastrointestinal inflammation known as necrotizing enterocolitis (NEC). NEC is characterized by exaggerated TLR4 signaling [71]. Normally, TLR4 activation induces a negative feedback loop that restricts its signaling, being mediated through the upregulation of the heat shock protein HSP70. HSP70 associates with TLR4 and recruits the ubiquitin E3 ligase CHIP, which then ubiquitinates TLR4 and targets it to proteasomal degradation [72]. Notably, pharmacological upregulation of HSP70 limited NEC by blocking TLR4-mediated inflammation, suggesting that modulating degradation may be utilized to overcome this fatal physiological condition [72]. Thus, there is great value in the identification of ubiquitination regulators and their relevant substrates, which may be utilized for developing therapeutic modalities for inflammatory bowel pathologies.

### 2.4. Proteasome Activity at the Immune Compartment of the Gut Barrier

While the previous examples describe the impact of proteasome activity in the gut epithelial cells on inflammation, various studies have demonstrated the involvement of proteasomal aberrations in immune cells during IBD. Below, we discuss several examples of proteasomal regulation in the adaptive and innate immune compartments of the gut.

Defective apoptosis of T cells at the lamina propria (LPT) is a hallmark of Crohn’s disease. Survivin, a member of the inhibitors of apoptosis family, was found to be stabilized in LPT of Crohn’s disease patients through increased interaction with HSP90 that protects it from proteasomal degradation [73].

Therefore, HSP90-dependent stabilization of survivin promotes T cell expansion and contributes to tissue inflammation. Interestingly, this phenomenon was found to be exclusive to Crohn’s disease but not colitis patients, suggestive of the different underlying mechanisms involved in these pathologies. Finally, an inhibitor of HSP90 could prevent T cell apoptosis, a feature that may be explored for therapeutic purposes in CD [73].

Signal transduction through TLRs plays a vital role in innate immune responses of myeloid cells in the gut by upregulating inflammatory cytokines through the transcription factors c-Rel and IRF5. The TLR signaling axis is negatively regulated by two TNFR-associated factor (TRAF) proteins, TRAF2 and TRAF3, whose specific depletion in myeloid cells promoted TLR3/4- induced inflammation and colitis in mouse models [49]. Mechanistically, TRAF2 and TRAF3 were demonstrated to associate with the E3 ligase cIAP, a member of the inhibitors of apoptosis family, to form an active E3 ligase complex that induced the K-48 linked polyubiquitination of c-Rel and IRF5 and their proteasomal degradation ([49], Figure 2B). Thus, TRAF2 and TRAF3, together with cIAP, have an anti-inflammatory role in the myeloid compartment of the gut by restricting TLR signaling. Interestingly, TRAF2 and TRA3 are upregulated in IBD [74]. Another example involves TRIM58-dependent degradation of TLR2 in myeloid cells [74]. Low levels of TRIM58 were identified in colons of ulcerative colitis patients, suggesting that restriction of innate immune activation by TRIM58-dependent TLR2 degradation is required to prevent excessive inflammation.

To conclude, while recent evidence suggests that regulation by ubiquitination and proteasome activity may be altered in IBD, the underlying mechanisms and the immuno-modulatory roles in shaping commensal and pathogenic bacteria at the gut are only starting to be elucidated. Furthermore, even from the limited knowledge we have gained, it is clear that numerous opportunities will emerge from manipulating and controlling the UPS systems to shape the epithelial barrier in the gut. In these, it will be critical to consider not only the clinical implications of reducing gut inflammation but also to maintain the homeostatic activities that are mediated by the UPS in the GI tract [58].

## 3. The UPS in Colorectal Cancer Development

Constitutive inflammation in patients suffering from IBD poses a significant risk for colorectal cancer (CRC), the third most frequently diagnosed cancer type worldwide [9,10,75,76,77]. CRC develops from adenomatous precursor lesions by a complex interaction of environmental factors along with the accumulation of genetic mutations. In addition, proteotoxicity has been attributed to contributing to CRC pathogenesis [78,79].

Given the critical role of proteasomes in maintaining cellular integrity, it is easy to imagine how altered expression or assembly of proteasome subunits may induce severe cellular aberrations and contribute to tumorigenesis. Indeed, the family of leucine zipper NRF transcription factors, which regulate proteasome expression, were highly implicated in colon cancer development. NRF1, which regulates proteasome expression, was shown to directly link metabolic reprogramming to proteostasis control in colorectal cells, via the UPS [80]. Specifically, in the basal state, NRF1 is targeted to degradation by the ubiquitin E3-ligase β-TrCP. Under conditions of high glycolytic and glutaminolysis activities, such as those prevalent in cancer cells, NRF1 becomes O-GlcNAcylated, which disrupts its association with β-TrCP, leading to NRF1 stabilization. In turn, the stabilized O-GlcNAcylated NRF1 promotes transcriptional activation of proteasome subunits, increasing proteolysis capabilities, which allows cells to prosper in the face of proteotoxic stress. It was further suggested that OGT-dependent upregulation of proteasomes contributes to the resistance of cancer cells to proteasome inhibitor treatment [80]. These findings suggest that addressing metabolic regulation of proteasomes in general, and by O-linked glycosylation in particular, may offer a novel understanding and avenues for sensitizing cancer cells to proteasome-inhibition therapy.

NRF2 expression in colon cancer was likewise suggested to induce proteasome expression and enable tumorigenesis by alleviating proteotoxic stress [81]. In this case, elevated expression of the proteasome subunits PSMA5 and PSMD4 were shown to protect from tumor necrosis factor-related apoptosis-inducing ligand (TRAIL)-induced apoptosis by enhancing NFκB activation [81]. In addition, NRF2 is induced by oxidative stress, and it was shown that NRF2 confers adaptation of colon epithelial cells to oxidative damage in samples from IBD. Thus, by enabling cells to cope with increasing stress conditions, NRF2 may promote tumor development and growth [82]. NRF3, which is highly expressed in various cancers, enhances 20S proteasomal assembly by inducing the expression of the 20S proteasome maturation protein POMP. In colorectal cancer, upregulation of POMP via NRF3 was shown to promote ubiquitin independent proteolysis of the tumor suppressors p53 and retinoblastoma, thereby promoting tumorigenesis and metastasis [83]. Notably, the significance of the NRF3–POMP axis was demonstrated in human colorectal and rectal cancer patients, wherein high expression levels of these regulators were correlated with poor prognosis [83]. Another factor affecting proteasome assembly in colon cancer is PSMD5, a non-ATPase subunit of the 26S proteasome, which acts as an inhibitor of proteasome assembly, leading to the accumulation of polyubiquitinated species in the cell. The expression of PSMD5 was found to be reduced in a human colon carcinoma cell line and in ex vivo mouse tumor organoids, likely contributing to cancer cell propagation by inducing proteasome activity [84].

Immunoproteasomes are highly expressed at the gut mucosa and have been linked to cancer through a plethora of cell-specific mechanisms, affecting both the epithelial cells and immune cells. Specifically, the expression of immunoproteasomes in epithelial cells affects the processing of antigens for presentation on MHC I, as well as the production of inflammatory cytokines and chemokines [85]. Further, immunoproteasomes regulate the activation of the NF-kB pathway [62] and the differentiation and activation of various adaptive and innate immune cells [86], thereby affecting the composite cellular environment of the gut mucosa.

Several studies have directly examined the involvement of immunoproteasomes in colon cancer. In one example, tumor initiation and progression were blocked by the immunoproteasome inhibitor ONX0914 in both chemically induced (AOM/DSS; [87]) and genetic (Apc^Min+^; [88]) mouse models of colon carcinogenesis. Immunoproteasome inhibition with ONX0914 reduced the overall tumor cell number and the CRC-associated loss of body weight and promoted the overall survival of mice. These effects were shown to be mediated by reducing the production of pro-tumorigenic chemokines such as CXCL1, CXCL2, and CXCL3 and pro-inflammatory cytokines such as IL-6 and TNF-α. Further, the block of NFκB activation by immunoproteasome inhibition abrogated the secretion of IL-17A, known to function as a carcinogen in the gut [88]. While proteasome inhibitors are the first line of cancer treatment in certain hematologic cancers, these results present the intriguing possibility to utilize selective inhibitors of the immunoproteasome in colon cancer.

Intriguingly, genetic polymorphism in LMP7 was identified in colorectal carcinoma patients as a susceptibility-associated allele. Specifically, the polymorphic allele was suggested to destabilize the LMP7 transcript, thereby hampering the upregulation of LMP7 under inflammation and reducing MHC presentation potentially through disrupting immunoproteasome assembly [89]. Thus, the effect of blocking immunoproteasomes in colon cancer should be carefully evaluated for the potential contrasting effects it may exert on tumor growth and anti-tumor immunity.

Deficiency of NLR family protein NLRP12 in non-hematopoietic cells majorly contributes to tumorigenesis by promoting non-canonical NFκB signaling. NLRP12 promotes the degradation of the NFκB, inducing kinase NIK, which thereby limits the activating proteolysis of NFκB. Further, NLRP12 associates with TRAF3, a negative regulator of non-canonical NFκB signaling. Depletion of NLRP12 reduced TRAF3 levels ([90], Figure 2B), Together these results indicate that NLRP12 functions as a critical checkpoint molecule associated with cancer-related inflammation via regulation of non-canonical NFκB signaling.

### Enzymes of the Ubiquitination Machinery in Colon Cancer

Several E3s and DUBs have been implicated in colon cancer (Figure 3). Among them, aberrations in the Wnt pathway are a hallmarks of CRC, observed in over 90% of patients, and various E3 ligases are implicated in the dysregulation of the Wnt pathway in CRC. Most CRC patients express a mutant form of the tumor suppressor gene adenomatous polyposis coli (APC), a regulator of the Wnt pathway, which is associated with polyp formation, an initiating stage of colon tumorigenesis. APC acts as a scaffold for the association of β-catenin with the destruction complex composed of Axin, CK1-α, and GSK-3β. Once associated with the complex, β-catenin is phosphorylated and ubiquitinated by the E3 ligase βTrCP, leading to its proteasomal degradation [91]. A mutation of APC in colon cancer creates a truncated form of the protein that is missing the domain responsible for the interaction with Axin. Therefore, the APC mutant disrupts the sequestration of β-catenin by the destruction complex, thereby increasing its cellular levels and the consequent expression of mitogenic genes such as Myc and cyclin D, contributing to the pathogenesis of colon cancer [91,92,93].

Additional frequent genetic alterations in colon cancer are the amplification of the E3 ligases RNF4 and RNF6 [94,95]. RNF4 is amplified in approximately 30% of colon adenocarcinoma patients. Ubiquitination of several oncoproteins including β-catenin, c-Myc, and c-Jun by RNF4 prevents their proteasomal degradation, thereby promoting their mitogenic functions [95]. RNF6 induces the ubiquitination and degradation of the transducin-like enhancer of split 3 (TLE3), a transcriptional repressor of the β-catenin/TCF4 complex. Thus, high RNF6 expression relieves β-catenin repression by TLE3 and is thereby the main contributor to colon carcinogenesis [94]. Further means by which the Wnt pathway is manipulated in cancer involve the stabilization of Wnt ligands such as Evi, whose levels are increased in colorectal cancer despite normal transcription, through regulation of degradation. Normally, Evi is targeted to degradation by the E2-conjugating enzyme UBE2J2 and the E3 ligase CGRRF1. Reduced expression of both of these enzymes is observed in colorectal cancer, leading to upregulation of Evi [96].

The zinc finger protein A20, also known as tumor necrosis factor-alpha-induced protein 3 (TNFAIP3), is a dual function enzyme that negatively regulates the NFκB pathway and suppresses TNF-induced apoptosis. A20 removes K63-linked ubiquitin modifications receptor-interacting protein 1 (RIP1) through its deubiquitinase activity and converts these to K48-linked polyubiquitination chains via its E3 ligase activity, leading to NFκB inactivation and suppression of the development of IBD. A20 binds to and ubiquitinates β-catenin, leading to its degradation and prevent colon tumorigenesis [97]. 

Additional ubiquitin E3 ligases have been described as tumor suppressors in the gut. The E3 ligase CHIP, often downregulated in CRC by promoter hypermethylation, targets the p65 subunit of the NFκB complex to degradation [98]. Consequently, the expression of NFκB-dependent genes that promote malignancy, such as cyclin D and c-Myc, is upregulated. Indeed, overexpression of cyclin D is a feature of various cancers including colon cancer. Interestingly, inactivating mutations in the Fbx4 E3 ligase, which directly promotes the degradation of cyclin D, are found in cancers overexpressing cyclin D. This exemplifies how mutations in specific E3s may serve to alter the stability of their downstream targets to promote carcinogenesis. Mechanistically, phosphorylation of Fbx4 on ser12 induces its association with 14-3-3ε, which promotes the Fbx4 E3 ligase activity [99], suggesting Fbx4 mutations, or 14-3-3ε expression, may serve as biomarkers for colon cancer.

The anaphase-promoting complex (APC/C) is a multi-subunit ubiquitin E3 ligase that functions as a checkpoint in cell cycle control, by targeting survivin, securin, and numerous mitotic substrates, including cyclins to degradation [100,101]. Dysregulation of APC function is associated with uncontrolled cell proliferation due to loss of cell cycle control. Interestingly, both loss and gain of function of the APC E3 ligase have been implicated in colorectal cancer. For example, loss of APC expression, frequently identified in colorectal cancer, is associated with poor prognosis [102]. However, overexpression of regulators that promote APC function can promote cancer as well. UbcH10, the E2 ubiquitin ligase of APC, is highly expressed in colorectal cancer and was suggested to act as an oncogene by promoting cell division through APC. Depletion of UbcH10 inhibited cancer cell proliferation and tumor growth in xenograft models, suggesting UbcH10 as a potential therapeutic target for colorectal cancer [103].

Beyond ubiquitination-promoting activities, deregulated expression or activity of deubiquitinating enzymes may disrupt degradation and proteostasis. The involvement of DUBs in colorectal cancer highlights the potential of utilizing DUB inhibitors as therapeutic options. The DUB inhibitor B-AP15 induces toxicity in colon cancer cells. While the specific substrates are not known, B-AP15 affects intracellular transport, influences aggresome formation, and enhances mitochondrial-dependent cytotoxicity [104]. Another DUB that has been associated with colorectal cancer is ubiquitin-specific protease 47 (USP47). USP47 upregulation in colorectal cancer was shown to prevent the proteasomal degradation of snail, an inhibitor of E-cadherin expression. This leads to the dissolution of cadherin-mediated cell-cell adhesion and reorganization of the cytoskeleton through epithelial–mesenchymal transition (EMT), thus acquiring migratory and invasive properties [105]. In addition, a recent study uncovered a role for USP11 in inducing mitogenic signals in colorectal cancer by stabilizing PPP1CA, an activator of the MAPK pathway [106]. USP7 was also shown to promote cancer by stabilizing β-catenin and inducing signaling regulating cell fate determination and migration via the Wnt pathway [107]. Specific inhibitors of USP7 (P5091 and parthenolide) were shown to attenuate Wnt/β-catenin-induced proliferation and migration and suppress tumor growth in HCT116 xenograft mouse models and are yet to be examined in clinical settings [108,109].

## 4. Concluding Remarks and Outlook

The gut epithelial barrier serves as an interface of microbiota, pathogens, and environmental antigens, highlighting the need for distinct sensing and response mechanisms that would be rapid and robust, yet highly specific and restrained. The diverse network of ubiquitination enzymes, including E2s and E3s that conjugate ubiquitin to substrates, as well as DUBs that remove ubiquitin conjugation, confer specificity for different substrates and functions. Compartmentalization of the ubiquitination enzymes and distinct types of proteasomes (i.e., constitutive vs. immuno) across the different cell types in the barrier tissue contributes to the ability of the gut to activate inflammation at the right place, at the right time, and for the right amount of time. For example, in certain cases of injury to the barrier integrity, inflammatory responses would be required in epithelial cells for tissue regeneration, yet immune responses should be limited to avoid excessive inflammation. Indeed, altered expression or activity of various ubiquitination enzymes has been implicated in oncogenic transformation in the gut. In other cases, disrupted ubiquitination and proteasome activity may lead to aberrant inflammation and is involved in the pathogenesis of IBD. It is therefore not surprising that various bacteria, both commensal and pathogenic, modulate proteasome expression and/or function to circumvent their propagation. Further alluding to the complexity of the UPS in the regulation of gut inflammation are accumulating evidence that describe the distinct patterns of UPS components in different gut pathologies. For example, the proteasome composition signature of epithelial cells is different in IBD patients from healthy individuals, but more strikingly, is different among IBD types such as ulcerative colitis and Crohn’s disease. These findings highlight the potential to consider proteasome composition and expression as a molecular profiling tool for diagnostics of IBD. However, how the integration of different degradation events is translated into determining a cellular inflammatory state entails identification of the specific substrates that are degraded under external inflammatory stimuli (e.g., infection, change in commensals), or under aberrant auto-inflammatory conditions. In this regard, a main challenge would be to untangle the complexity and specificity of the ubiquitin E3 ligase network and the different proteasome complexes that function at the gut, including their tissue-specific functions and their modulation by host-driven or pathogen-driven cues. Further characterization of the degradation landscape [110,111] in IBD may also shed new light on yet undiscovered proteolysis-dependent mechanisms. Such new layers of information into the physiology and pathophysiology of gut inflammation will offer novel molecular profiling approaches and new treatment possibilities in IBD and colorectal cancer.

Beyond the gut barrier, the involvement of the UPS in other barrier tissues may be also relevant to a different aspect of innate immunity and signaling control. With the advent of proteolysis-targeting chimeras (PROTACs; [112,113,114], it is tantalizing to think that a better understanding of UPS-mediated regulation may be adopted to reshape the cellular environment in a controlled manner to regain tissue homeostasis.

## Figures and Tables

**Figure 1 biomolecules-11-00989-f001:**
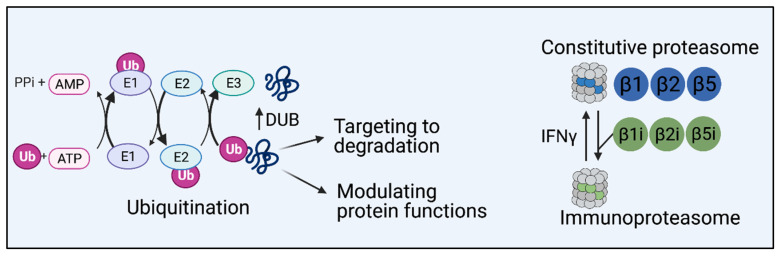
**Zoom-in on ubiquitin-dependent protein degradation**: Ubiquitin-dependent protein degradation is a multi-step process. Ubiquitin is loaded onto an E1-activating enzyme and transferred to an E2 conjugating enzyme. Ubiquitin E3 ligases recognize specific substrates and facilitate the conjugation of ubiquitination. Ubiquitinated proteins are then recognized by the 26S proteasome for degradation. The proteolytic subunits of constitutive proteasome are β1 (PSMB6), β2 (PSMB7), and β5 (PSMB5). IFNγ stimulates the formation of the immunoproteasome, which is composed of alternative catalytic subunits, namely, β1i (PSMB9 or LMP2), β2i (PSMB10 or MECL-1), and β5i (PSMB8 or LMP7) in turn of their constitutive counterparts. Substrates are deubiquitinated at the proteasome to allow for recycling of ubiquitin molecules and are unfolded and translocated into the proteasome where they are cleaved into peptides by the catalytic subunits [25,26,38,39,40].

**Figure 2 biomolecules-11-00989-f002:**
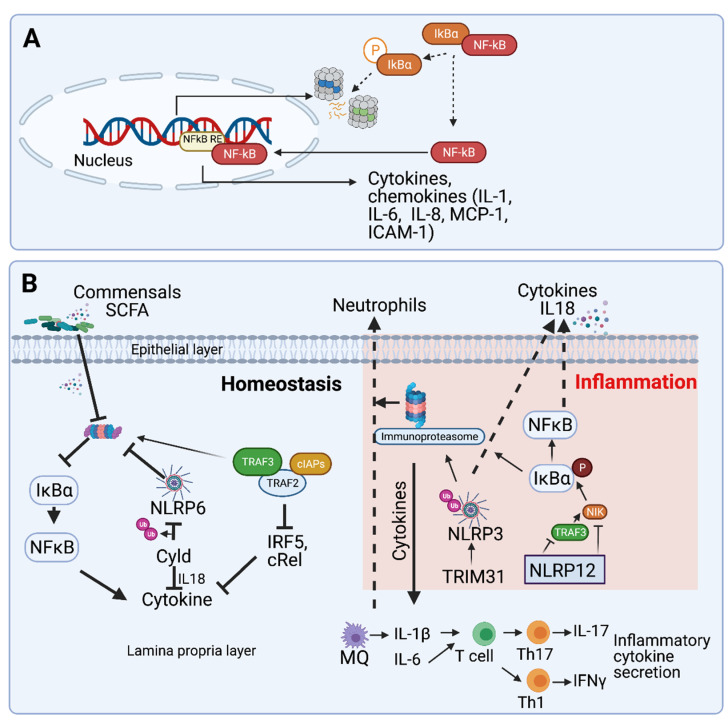
(**A**) **Proteasome-dependent NFκB signaling:** Under steady state, NFκB is kept inactive by association with IκB. Inflammatory signals induce the phosphorylation of IκB, its dissociation from NFκB, and degradation by the proteasome. Upon the release from IκB, NFκB translocates to the nucleus, leading to transactivation of inflammatory cytokines, chemokines (IL-6, IL-8, IL-1, etc.), and apoptotic genes (ICAM-1, Bcl) [23]. (**B**) **Proteasomes at the crossroad between gastrointestinal homeostasis and inflammation:** Various proteasomal degradation events restrict constitutive inflammation and maintain homeostasis. Metabolites secreted by commensals such as *L. casei* block proteasome activation, which in turn leads to reduction in IκB degradation, thereby hindering NFκB activation [47]. Cyld, a deubiquitinating enzyme, deubiquitinates NLRP6, prevents NLRP6 degradation, and enhances release of IL-18 [48]. The TRAF2/3 complex activates cRel and IRF5 that induce pro-inflammatory cytokines in response to danger stimuli. The E3 ligase cIAP is recruited to the TRAF2/3 complex, leading to the degradation of both TRAF2 and TRAF3, thereby limiting production of inflammatory cytokines [49].

**Figure 3 biomolecules-11-00989-f003:**
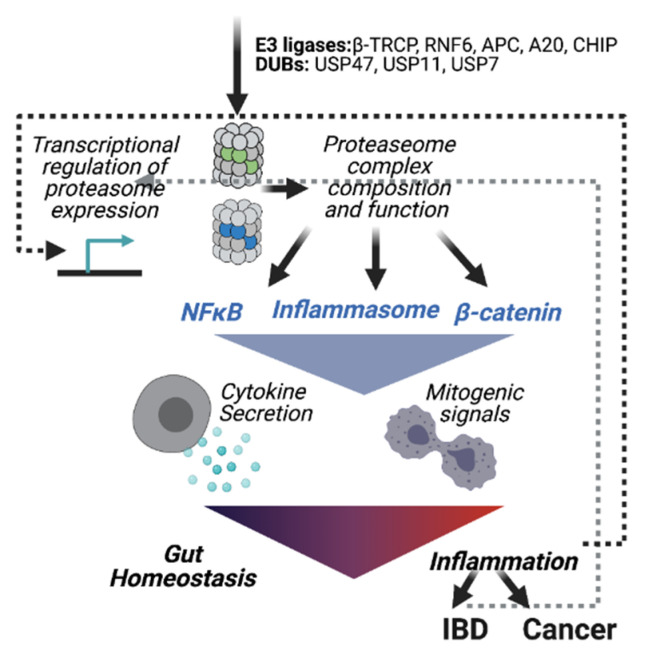
**Snapshot of ubiquitin proteasome-dependent gastrointestinal inflammation and colorectal cancer.** Gut homeostasis is regulated via the function of various ubiquitination enzymes (E3 ligases and DUBs) that regulate NFκB signaling, inflammasome activation, and β-catenin signaling. Altered expression or functions of ubiquitination enzymes, or proteasome subunits, is associated aberrant production of pro-inflammatory cytokines and immune cell differentiation, which disrupts homeostasis and promotes gut inflammation, potentially inducing chronic inflammatory bowel disease or colorectal cancer.

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
