# Peer review of "Gatekeepers of the Gut: The Roles of Proteasomes at the Gastrointestinal Barrier"

_biomolecules, 2021, doi:10.3390/biom11070989_

Round 1
Reviewer 1 Report
The Ubiquitin-Proteasome System at the Firewall: Roles at the Gastrointestinal Barrier
Mohapatra et al
The authors have reviewed proteasome-dependent regulation of molecular processes in the gut in the context of inflammatory disorders and tumorigenesis. The topic is relevant and current, but I have some comments on the presentation of the theme.
While the review is well written and informative, it is not comprehensive and the selection of the particular effectors should be justified in a couple of sentences. The text is otherwise easy to read, but some general conclusions are repeated very many times. The selected division of information into sections is a bit hard to follow, as the authors jump from one theme to another. I also would like the authors to be more clear about this review describing effects on proteasomal degradation that is part of the inflammatory cell signalling in the intestine. It should be clear also to a reader that is not an expert in the field that this is not the same thing as changes in certain proteins (for example by mutation) that affects the stability and hence leads to changes in proteasomal degradation.
The biggest problem of the review are the figures. They are not of good quality, are not informative, and does not support the text.
Figure 1.
- The figure title does not correspond to the figure.
- The figure is referred to in a strange context difficult to interpret “Ultimately, controlling the equilibrium between the two potential consequences of inflammation, namely cytoprotection, the prevention and healing of ulcers, vs. injury, dictates the physiological outcome of the response. Indeed, modulation of proteasome-dependent NF-kappaB activation impacts gut homeostasis by maintaining the balance of gut inflammation (Figure 1)”. These sentences should be revised and the figure should be referred to at the sites in the text that correspond to the figure.
- The figure legend needs to be rewritten, it is a repetition of the text and all parts are not included in the figure. The figure legend should describe what is shown in the figure, not all kinds of other things.
- The arrow with deubiquitinase from the proteosome is confusing, as the deubiquitination happens before proteins are processed in the proteaosome.
- The illustration of the immunoproteasomes is not informative, and it should be referred to in the part describing the immunoproteasome.
Figure 2.
- The figure does not correspond to the figure legend and is not referred to correctly in the text.
- The legend describing B does not match at all with the figure, sections A and B seem to be in the wrong place and the C-part is described in legend but not shown in the figure. The arrow from IkBa to the nucleus in the first part of the figure does not makes sense.
- The figure should be referred to already in the section: “Ubiquitin E3 ligases in inflammatory gut disease”.
- The quality of the image needs to be improved as some of the text is unreadable.
- As such the figure does not serve a purpose in the review.
The box on the UPS-system:
- It is stated that ubiquitin chains are made by lysine residues that may be conjugated by other ubiquitin molecules. Also the N-terminal methionine can be used for conjugation, and this should be mentioned.
- It is also mentioned that proteasomal degradation is most associated with K48-linked chains. This was certainly true in 2012, which is also the year of the review used as base for this statement, but the authors have to mention the current knowledge on K11/K48 branched chains being the major determinants of protein degradation. The authors should base their claims on recent literature.
- A section about DUBs should be added.
- Figure 1a should be connected to the text in the box.
Other comments:
- The authors describe that commensals inhibit proteasome activity to inhibit inflammation. They should also add some information about the molecular consequences of this inhibition.
- A paragraph regarding the role of non-canonical NF-?B NIK on intestinal inflammation could be added. NLRP12 is shown to promote proteasomal degradation of NIK and NLRP12-/- mice are highly susceptible to colitis and colitis-associated colorectal cancer.
- In addition to CYLD, A20 could be mentioned as a regulator of intestinal inflammation.
- It would be nice with a separate section for therapeutic targets for gastrointestinal UPS, where the therapeutic targets would be described.
- The section “The need for a rheostat of inflammation” seems like a second introduction and could be baked into the introduction.
- The section “The UPS in CRC development” is very short and could be combined with “Factors influencing proteasome expression and assembly in colon cancer”.
- The title is not linguistically good.
- Crohn’s disease, not Chron’s disease
- The authors should use Greek letters in names such as NF-kappaB, IkappaBalpha and IFNgamma throughout the manuscript, also in figures and figure legends, and adhere to one style of writing the names of these proteins and spell out the abbreviation the first time used.
Reviewer 2 Report
The manuscript by Mohapatra et al. analyzes the current knowledge about the role of the ubiquitin-proteasome system in the control of the intestinal epithelium during inflammation and cancer. The work is well written and structured. It is easy to follow and very comprehensive. Given the potential of the UPS system as a target to develop treatments this review is very timely and can offer good insights into new applications of the UPS modulators in intestinal diseases. I find the review interesting and thorough and I think it is ready for publication. I have a couple of comments/changes I would like the authors to consider:
- Figure 1 shows the E3 ubiquitin ligase attached to Ub when many E3 ligases do not work this way. I feel this is misleading and could be portrayed in a different way.
- No reference is provided for the statement that: "It was further suggested that these commensal bacteria induce selective interference with proteasomal degradation of specific substrates..." I think this is relevant enough to show the related references.
- In this same section when talking about CYLD the authors could mention the recent paper by Mukherjee et al in Nat Immunol regarding the regulation of NLRP6 by this deubiquitinase.
- In the section regarding the proteasome in inflammation it could be useful to develop the concept of balance between positive and negative regulation of the inflammation that is introduced in the last paragraph of the section.
- The recent report regarding the control of AKT by RNF8 in ulcerative colitis (Zhu et al 2020) could be included in the section of E3 ubiquitin ligases in inflammatory disease.
- The authors could discuss the emerging role of SUMO in the control of the intestinal epithelium and cancer development.
Reviewer 3 Report
The review aims to describe the role of the ubiquitin-proteasome system (UPS) in maintaining gut homeostasis and in the pathogenesis of inflammatory bowel disease and colon cancer.
Overall, the review is thorough an informative. It could be improved with a better organization of the manuscript. The introduction in numbered 1 but there is no 2 or 3. There is 2 major parts (inflammatory disease and cancer) that should be separated.
The first part concerning the inflammatory pathologies is quite difficult to follow. It could be interesting to introduce the different kinds of gut inflammatory diseases. The authors should explain the function of the different regulators of the UPS described in this part: for example, what is the role of HSP, STING, survivin?
The part on TLRs in the colon myeloid compartment in confusing. 1) TLR stimulation upregulates c-Rel and IRF5 which induce the expression of inflammatory cytokines. 2) TRAF2 and TRAF3 induce the degradation of IRF5 and c-Rel. Then, the upregulation of TRAF2 and TRAF3 should lead to a downregulation of c-REL and IRF5 and therefore a decreased level of inflammatory cytokines.
The Figure 2 legend did not explain the first part of the Figure (in blue) and there is no C in the Figure.
Round 2
Reviewer 1 Report
Gatekeepers of the Gut: The roles of proteasomes at the gastrointestinal barrier
Mohapatra et al
The authors have done a thorough job in improving the manuscript, and have sufficiently responded to my comments. However, the manuscript still contain many typos and needs a through spellcheck, for example the spelling of Crohn’s disease is still not corrected in several places. While it is good that the authors refer to older studies describing original research, some of the review articles referred to are almost 10 years old and could be exchanged to more current ones. The figure legends also lack references. In addition, I would suggest that Figure 1 A would be incorporated in the text box describing ubiquitination.
Reviewer 3 Report
No comment
Author Response
There was no comment from the reviewer.